# Should I Stay or Should I Go? Associations between Occupational Factors, Signs of Exhaustion, and the Intention to Change Workplace among Swedish Principals

**DOI:** 10.3390/ijerph18105376

**Published:** 2021-05-18

**Authors:** Inger Arvidsson, Ulf Leo, Anna Oudin, Kerstin Nilsson, Carita Håkansson, Kai Österberg, Roger Persson

**Affiliations:** 1Division of Occupational and Environmental Medicine, Lund University, SE-223 81 Lund, Sweden; anna.oudin@med.lu.se (A.O.); kerstin.nilsson@med.lu.se (K.N.); carita.hakansson@med.lu.se (C.H.); roger.persson@psy.lu.se (R.P.); 2Centre for Principal Development, Umeå University, SE-901 87 Umeå, Sweden; ulf.leo@umu.se; 3Department of Public Health and Clinical Medicine, Umeå University, SE-901 87 Umeå, Sweden; 4Department of Public Health, Kristianstad University, SE-291 88 Kristianstad, Sweden; 5Department of Psychology, Lund University, SE-221 00 Lund, Sweden; kai.osterberg@psy.lu.se

**Keywords:** psychosocial working conditions, mental health, school leader

## Abstract

A high turnover among principals may disrupt the continuity of leadership and negatively affect teachers and, by extension, the students. The aim was to investigate to what extent various work environment factors and signs of exhaustion were associated with reported intentions to change workplace among principals working in compulsory schools. A web-based questionnaire was administered twice, in 2018 and in 2019. Part I of the study involved cross-sectional analyses of the associations 2018 (n = 984) and 2019 (n = 884) between occupational factors, signs of exhaustion, and the intention to change workplace, using Generalized Estimating Equations models. Part II involved 631 principals who participated in both surveys. The patterns of intended and actual changes of workplace across two years were described, together with associated changes of occupational factors and signs of exhaustion. Supportive management was associated with an intention to stay, while demanding role conflicts and the feeling of being squeezed between management and co-workers (buffer-function) were associated with the intention to change workplace. The principals who intended to change their workplace reported more signs of exhaustion. To increase retention among principals, systematic efforts are probably needed at the national, municipal, and local level, in order to improve their working conditions.

## 1. Introduction

The leadership of principals is of great importance for the school. The reports of a high turnover among principals are worrying [1,2,3,4], since it may disrupt the continuity of the leadership and negatively impact the school climate [5]. Long-term relationships are required to build mutual trust within the organization [6,7], which forms the basis for being able to offer support and adequate development opportunities for teachers [8]. It has been observed that frequent changes of principals results in lower teacher retention, which is particularly harmful for low-achieving schools with many inexperienced teachers [1]. Moreover, associations between principal turnover and student performance have been observed [1,2,3]. In a Swedish context, the National Agency for Education [9] and the Swedish schools-inspectorate [10] have expressed concerns that a high principal turnover may impair opportunities for students to achieve their goals. According to The Teaching and Learning International Survey (TALIS) [11], the median number of years working at their current school was three years for Swedish principals. Corresponding values in the Nordic countries ranged between four and six years, and the median value was five years across all 48 countries participating in TALIS 2018. It thus seems as if a problematic principal turnover occurs to a large extent in Sweden and it is therefore important to improve the understanding of its underlying factors. 

There may be a variety of reasons why principals decide to change workplaces. One reason may be that the change represents a natural career move, for example, from being an assistant principal to principal. Other reasons may have more practical and private origins. Anyhow, complex and demanding working conditions among principals have been reported [12,13,14,15], which may be of importance for the principals’ inclination to stay or change workplaces. In a review of principal-turnover research [2], it was pointed out that inadequate preparation and professional development, poor working conditions, and a lack of decision-making authority were some of the reasons why principals leave their jobs. In addition, in Sweden, large variations in principal turnover have been observed between schools and municipalities [3], as well as between different school forms [10]. These indicate variable conditions depending on both external factors (e.g., socioeconomic factors) and the working conditions within the organizations. 

While work environment factors and stress-related health among teachers have gained a large interest among researchers [16,17,18,19,20], less scientific studies have focused on the work environment and health among principals [13,21,22]. Anyhow, evidence has been provided of an association between principals’ satisfaction with their work characteristics and their mental health status [12]. Furthermore, several sources of occupational stress have been identified, for example work overload, handling relationships with staff, teacher climate, lack of resources and personal characteristics [13,23,24]. As a leader of the school, the principals’ work is associated with a wide range of responsibilities and roles comprising pedagogical development, staffing, finances, premises, and work environment issues, to name a few. Furthermore, the principal may experience numerous expectations from the school owner, superintendents, the schools-inspectorate, co-workers, parents, and students [14]. Whether these demanding organizational and social work environment factors are balanced by supportive management, supportive manager colleagues, or organizational structures, is most likely of great importance for their well-being and intention to stay in the organization [25]. A previous literature review of studies in different occupational groups has shown that risk factors such as high demands, low job control, and a high workload, increased the risk for exhaustion and poor mental health [26]. The association between burnout and high job demands and low job control also applied to Swedish teachers [19,20]. In a previous study of the present study population [22], 29% of the principals and assistant principals in compulsory school met the exhaustion criteria according to Karolinska Exhaustion Disorder Scale (KEDS) [27]. This prevalence rate is almost twice as large as the prevalence rate observed in a highly educated but occupationally diverse study sample [28]. Since a clear association has been found between a high level of stress and the intention to change jobs among teachers [29], it is of interest to find out whether this is true also among principals. Furthermore, to gain knowledge for preventive actions, there is a need to identify which specific occupational factors are of importance for the principals’ intention to stay at, or leave, their current workplace. 

### Aims

The aims of the present study were twofold. The first aim (Part I) was to identify which occupational factors (organizational structures and work environmental factors) were associated with the intention to stay or the intention to change workplace, among principals and assistant principals in Swedish compulsory schools. The second aim (Part II) was to describe the patterns of intended and actual change of workplace across two years (2018 and 2019), and the associations with occupational factors and signs of exhaustion. 

## 2. Materials and Methods

### 2.1. Study Design

The present study was a part of a longitudinal study of working conditions and health among principals in 277 municipalities in Sweden that included two web-based questionnaires using the software Textalk Textalk (Mölndal/Gothenburg, Sweden; www.textalk.com/products/websurvey/ accessed on 11 November 2019). The questionnaires were administered with a one-year interval in September/October 2018 and in September/October 2019. 

In the present study, both cross-sectional and longitudinal analyses were performed, and the study comprised two parts. Part I involved cross-sectional analyses of the associations of year 2018 (n = 984) and year 2019 (n = 884) between occupational factors, signs of exhaustion, and the intention to change workplace, among principals and assistant principals working in compulsory schools. The cross-sectional analyses accounted for the fact that some participants had responded to the survey on both occasions and thus had repeated measurements (see statistical analyses). 

Part II was a longitudinal analysis comprising 631 principals and assistant principals who participated in both surveys and who worked in compulsory schools in 2018. The patterns of intended and actual changes of workplace across two years (2018 and 2019) were described. Analyses were performed of associations between changes of the intention to stay, or leave the current workplace between 2018 and 2019, and changes of occupational factors and signs of exhaustion during the same period (with all actual workplace changes taken into account).

### 2.2. Participants

The participants in the longitudinal study were recruited via e-mail address by principals who had participated in training programs for principals funded and arranged by the Swedish National Agency for Education, during the period 2008–2017. In 2018, 9900 principals potentially working in pre-schools, compulsory schools, upper secondary schools, or adult education were invited to participate in the study. Of these, 4640 individuals either accepted (n = 2633) or declined (n = 2007) participation. Eventually, 2317 completed the entire questionnaire in 2018. The response rate was 50%, based on the principals that responded yes and also completed the survey, versus no-responders and those that did not complete the survey. In 2019, the second questionnaire was directed to the 2316 presumed still occupationally active responders in the first survey, of which 1528 responded. Furthermore, the participants who did not answer in the first assessment round (in 2018) were re-invited to the second round. Out of the 5149 previously silent individuals, 3350 were seemingly reached without technical problems (i.e., bouncing e-mail), of which 464 decided to complete the second questionnaire. Thus, the second survey comprised a total of 1992 principals. The study was approved by the Regional Ethical Review Board in Lund, Sweden, Dnr 2018/247.

Part I included principals and assistant principals working in compulsory schools, which comprised 1000 out of the 2316 respondents in 2018, and 901 out of the 1992 respondents in 2019. Participants who stated “other title” were omitted from the study (n = 15 in both years 2018 and 2019). Due to internal missing values, another three individuals (one in 2018 and two in 2019) were omitted. Thus, the present study sample comprised 984 principals (720 women and 264 men; mean age 49; range 30–67 years old) who responded to the first questionnaire and 884 principals (649 women and 235 men) who responded to the second one. Among these, 633 responded to both questionnaires. In total, 1235 principals had completed at least one of the questionnaires. 

Part II involved the principals and assistant principals who participated in both surveys and worked in compulsory schools in 2018, irrespective of any change to another school-level in 2019 (n = 651). Twenty participants who stated that they would retire in 2018 or 2019 were excluded from the study, resulting in a study sample of 631 individuals.

### 2.3. Outcome Measure

In Part I, the outcome of the study was based on the question “Do you intend to change workplace within the coming two years?” with the response alternative 1: Yes definitely; 2: Yes, probably; 3: No; and 4: I will retire. The dichotomous outcome consisted of 1: No (i.e., intention to stay) vs. 2: yes, probably/yes, definitely (i.e., intention to leave). Participants who responded that they would retire were omitted.

### 2.4. Independent Factors

The data collection, for both Part I and Part II, comprised questions on gender, age, and occupational factors, i.e., job title (principal/assistant principal), school owner (municipality/other organization), seniority as principal, the number of co-workers and the number of students the principal was responsible for. Furthermore, information was collected on the staff availability (1: Full staff or more than full staff; 2: somewhat or very understaffed), overtime work (1: Once a week or less often, 2: a few times a week or every day; 3: No agreed working hours), and perceived physical working environment (1: Adequate, good or very good; 2: poor or very poor).

Demanding and Supportive organizational and social work environment factors were measured with Gothenburg Manager Stress Inventory (GMSI). GMSI-mini is a brief version of GMSI [30], which has been has been shortened by one of the authors (KÖ), in consultation with the developers Mats Eklöf and Anders Pousette. The selected items showed the highest correlation with the corresponding subscale in the original GMSI, which have been analyzed in two different samples. Demanding work environment factors were assessed in relation to the preceding six months, by 22 items distributed along eight dimensions, i.e., Resource deficits, Organizational control, Role conflicts, Role demands, Group dynamics, Buffer-function, Co-workers, and Container-function (i.e., dealing with co-workers’ problems and frustrations). In each dimension, all questions were answered along a five-point scale with the options 1: “never/almost never”; 2: “Rarely”; 3: “Sometimes”; 4: “Often” and 5: “Always/almost always”. The supportive organizational and social work environment factors were assessed with 10 items distributed along five dimensions, i.e., Supportive management, Cooperating co-workers, Supportive manager colleagues, Supportive private life, and Supportive organizational structures. The responses to each item were given along a five-point scale, indicating the level of agreement with various statements about supportive factors; 1: “Applies very poorly”, 2: “Applies poorly”, 3: “Applies to some extent”, 4: “Applies well”, and 5: “Applies very well”. For both demanding and supportive factors, the mean score in each dimension was calculated for each individual. A higher mean score indicated more frequent experience with actual demands, and the perception of better support. In Appendix A, the number of items and a description of the included questions are given for each GMSI-dimension (this information has previously been presented in [15]). 

In study specific questions, the participants were asked about “To what extent do you perceive stressful external expectations” from the following eight actors: the Swedish schools-inspectorate, the National Agency for Education, the school owner, Superintendent, immediate manager, co-workers, parents, and students. Each of the eight questions was answered along a seven-point scale ranging from 0 (verbalized as “not stressful at all”) to 6 (very stressful). Thereafter, in Part I of the study, the scales were dichotomized into low (<4 points) and high (≥4 points) stressful external expectations. In Part II, the mean value on each scale was calculated for comparisons within and between specific groups of principals (see further description below). 

Signs of exhaustion were measured with The Karolinska Exhaustion Disorder Scale (KEDS) [27], which includes 9 items that refer to the last 2 weeks, comprising the following domains: (1) ability to concentrate, (2) memory, (3) physical resistance, (4) mental resistance, (5) recovery, (6) sleep, (7) hypersensitivity to sensory impressions, (8) experience with demands, and (9) irritation and anger. Each item was answered on a 7-point scale (0–6). The sum score was calculated for each individual (0–54 possible) and higher values reflected more severe symptoms. The mean value, as well as the prevalence of a score ≥ 19 points, which indicated signs of exhaustion/possible exhaustion disorder [25], was calculated for specific groups of principals. 

The participants were also asked, “How many times have you changed workplace during the past five years?” (2018) and “How many times have you changed workplace during the past 12 months” (2019), with the response alternatives “none”, “once”, “twice”, or “three times or more”. The latter question was used in Part II of the study where “None” indicated no change of workplace (=No), and once or more indicated that the principal had changed workplace (=Yes). 

In Table 1, descriptive estimates for the occupational factors, signs of exhaustion, and intended and actual changes of workplace are presented for 2018 and 2019, for the participants included in Part I of the study.

### 2.5. Part II: Longitudinal Patterns of Intended and Actual Change of Workplace

The patterns of intention to change workplace in the year 2018 and year 2019 (no vs. yes, probably/yes, definitely), together with the actual change of workplace reported in 2019, are described in Figure 1.

A large proportion of the principals remained at their workplace and had no intention of changing workplace, neither in 2018 nor in 2019. However, some principals had changed their mind concerning the intention of changing, or staying at the present workplace between 2018 and 2019. This may of course be due to the fact that they had actually changed workplace during the year, and of natural reasons had other working conditions. However, the reason for them to change their mind may also be due to changes of their working conditions and/or stress-related health at the present workplace, which is of interest to study. 

To distinguish the groups, the participants were first divided into two groups based on the reports in 2018: those who intended to stay and those who intended to change workplace within the next two years. In the next step, the two groups were divided into four, based on the question in the survey in 2019 “Have you changed workplace during the past twelve months?” (No vs. Yes). In the third step, these four groups were divided into 8 groups, based on their intention to stay or leave their current workplace in 2019. This revealed eight different paths: 1.Intention to stay 2018—No change of workplace—Intention to stay 2019(Stay/No change/Stay)2.Intention to stay 2018—No change of workplace—Intention to leave 2019(Stay/No change/Leave)3.Intention to stay 2018—Change of workplace—Intention to stay 2019(Stay/Change/Stay)4.Intention to stay 2018—Change of workplace—Intention to leave 2019Stay/Change/Leave)5.Intention to leave 2018—No change of workplace—Intention to stay 2019(Leave/No change/Stay)6.Intention to leave 2018—No change of workplace—Intention to leave 2019(Leave/No change/Leave)7.Intention to leave 2018—Change of workplace—Intention to stay 2019(Leave/Change/Stay)8.Intention to leave 2018—Change of workplace—Intention to leave 2019(Leave/Change/Leave)

As an indicator of stress-related health in each of the groups, the prevalence of a sum score ≥ 19 points (signs of exhaustion; KEDS [24]) is shown in Figure 1.

For in-depth analyses of factors that may be of importance for the intention to change workplace, we chose to perform pairwise comparisons of occupational factors and signs of exhaustion in years 2019 and 2018, within and between Group 1 (Stay/No change/Stay) and Group 2 (Stay/No change/Leave), within and between Group 5 (Leave/No change/Stay) and Group 6 (Leave/No change/Leave), and within and between Group 7 (Leave/Change/Stay) and Group 8 (Leave/Change/Leave) (Appendix A). The Groups 3 (Stay/Change/Stay) and 4 (Stay/Change/Leave) were not involved in these in-depth analyses, because they comprised a limited number of participants (N = 16 and N = 4, respectively). Longitudinal patterns of intended and actual change of workplace are shown in Figure 1, which is placed further ahead in the results (Section 3.2).

### 2.6. Statistical Analyses

All statistical analyses were performed with IBM SPSS software, version 24 (IBM Corp.). *p*-values ≤ 0.05 (two-tailed) were considered statistically significant.

To estimate the associations between gender, age, occupational factors, and signs of exhaustion, and the intention to change workplace (No vs. yes, probably/yes, definitely) Generalised Estimating Equation (GEE) models were used to specify a repeated measure model with a dichotomous outcome (binary logistic regression). Thus, survey data from 2018 were combined with survey data from 2019, taking into account that 633 out of 1235 study subjects had participated in both surveys (dependent observations). The intention to change workplace was first estimated in bivariate GEE models for all variables (gender, age, occupational factors, and signs of exhaustion) by Odds ratios (ORs) and their 95% confidence intervals (CIs). In the next step, ORs for intention to change workplace were estimated using multivariate GEE, for all variables with bivariate *p*-values < 0.1. In the multivariate GEE model, five variables concerning stressful external expectations were omitted from the analyses (i.e., expectations from the Swedish schools-inspectorate, National Agency of Education, school owner, superintendent, and immediate supervisor), due to a conceptual overlap with the variables in GMSI regarding demanding organizational and social work environment factors.

In Part II of the study, analysis within the groups (2019 vs. 2018) were done using the Wilcoxon Signed Rank Test for continuous variables and McNemar’s test for dichotomous variables. Comparisons between the groups were performed using the nonparametric Mann–Whitney U-test for continuous variables and Fisher’s exact test for dichotomous variables.

## 3. Results

### 3.1. Part I

In 2018, 98 principals out of 984 (10%) reported that they definitely intended to change workplace within the coming two years, and 304 (31%) that they probably would do so (Table 1). Corresponding values for the 884 participants in 2019 were 96 (11%) and 271 (31%), respectively.

#### 3.1.1. Bivariate Analyses of Associations between Occupational Factors, Gender, Age, and Signs of Exhaustion, and the Intention to Change Workplace

Younger age, assistant principal, understaffing, and a perceived poor physical working environment were all associated with an intention to change workplace (Table 2). In addition, for all of the demanding organizational and social work environment factors in GMSI, high scores were statistically significantly associated with the intention to change workplace. The same was true for stressful external expectations from the school owner, superintendent, immediate manager, co-workers, and parents. Furthermore, a high sum-score on the exhaustion scale (KEDS) was associated with the intention to change workplace.

For all of the supportive organizational and social work environment factors, except for a Supportive private life, high scores were significantly associated with the intention to stay at the current workplace (Table 2). There were no statistically significant associations between the intention to change workplace, and gender, school owner, seniority, number of students and co-workers, and overtime work.

#### 3.1.2. Multivariate Analyses of Associations between Occupational Factors, Age, and Signs of Exhaustion, and the Intention to Change Workplace

In the multivariate analyses, younger age and assistant principal remained statistically significantly associated with the intention to change workplace (Table 2). The same was true for high scores of stressful external expectations from parents. Among the demanding organizational and social work environment factors (GMSI), Role conflicts, (OR 1.22 (CI 1.03–1.46), associated with a one unit increase in mean score, on the scale 1–5) and Buffer-function, OR 1.27 (CI 1.08–1.50) remained statistically significant, while Resource deficits, Organizational Control, Role demands, Group dynamics, Co-workers and Container-function did not. Furthermore, a high score of a Supportive private life was associated with the intention to change workplace. The strongest association with an intention to change workplace was found for a high sum-score on the exhaustion scale (KEDS) (OR 1.03 (CI 1.02–1.05), associated with a one unit increase in mean score, on the scale 0–54].

High scores of Supportive management (OR 0.79 (CI 0.71–0.89), associated with a one unit decrease in mean score, on the scale 1–5) and Supportive manager colleagues remained statistically significantly associated with the intention to stay at the current workplace, while Cooperating with co-workers and Supportive organizational structures did not (Table 2). 

### 3.2. Part II: Longitudinal Patterns of Intended and Actual Change of Workplace

As described in the methods section, and illustrated in Figure 1, the longitudinal analysis revealed eight different paths of intended and/or actual changes of workplace during 2018 and 2019.

#### 3.2.1. Group 1 (Stay/No Change/Stay) and Group 2 (Stay/No Change/Leave)

The principals in Group 1 (N = 274) had no intention to change workplace neither in 2018 nor in 2019, and reported no change of workplace in the survey in 2019 (Figure 1). 

The longitudinal results within the group showed generally better working conditions in 2019 compared to 2018 (Table 3). 

The principals in Group 2 (N = 99) had no intention to change workplace in 2018. In 2019, no actual change of workplace was reported, but they now reported that they had the intention to change workplace within the next two years (Figure 1). The longitudinal results within Group 2 showed statistically significantly impaired scores in 2019 compared to 2018, in terms of Resource deficits, Buffer-functions, and Supportive management (Table 3).

Differences between Group 1 and Group 2: Already in 2018, the principals in Group 1 reported generally better working conditions than the principals in Group 2 (Table 3). In 2019, most of the differences were accentuated. Group 2 reported statistically significantly higher scores in external expectations from school owners, immediate supervisors and co-workers, and in all of the demanding organizational dimensions. Group 2 also reported less Supportive Management and less Supportive organizational resources than Group 1. A higher proportion of the principals in Group 1 were employed in an organization other than in the municipality and they were responsible for a higher number of students. Group 2 reported significantly higher scores on the exhaustion scale (KEDS) in 2018, and the difference between the groups was accentuated in 2019. 

#### 3.2.2. Group 5 (Leave/No Change/Stay) and Group 6 (Leave/No Change/Leave)

In 2018, the principals in Group 5 (N = 52) reported that they had the intention to change workplace within the next two years. In 2019, no actual change of workplace was reported, and they no longer had the intention to change workplace (Figure 1). In 2019, they reported generally better working conditions compared to 2018 (Table 4). The scores of stressful external expectations from the superintendent and the immediate supervisor declined, and the same was true for Role conflicts and Container functions. Group 5 also reported statistically significantly lower scores on the exhaustion scale (KEDS) in 2019 compared to 2018. 

The principals in Group 6 (N = 146) reported both in 2018 and in 2019 that they had the intention to change workplace within the next two years, but no actual change of workplace was reported in 2019 (Figure 1). Within the group, there were only minor changes of the reported working conditions and signs of exhaustion in 2019 compared to 2018 (Table 4). 

Differences between Group 5 and Group 6: In the survey in 2018, there were no statistically significant differences between Group 5 and Group 6, neither in occupational factors nor in exhaustion (Table 4). In 2019, the principals in Group 5 reported better conditions with regards to stressful external expectations from the immediate supervisor and co-workers, and lower scores regarding Role conflicts and Co-workers’ problems, compared to Group 6. Furthermore, in 2019, Group 5 reported significantly lower scores on the KEDS exhaustions scale, compared to Group 6.

**Figure 1 ijerph-18-05376-f001:**
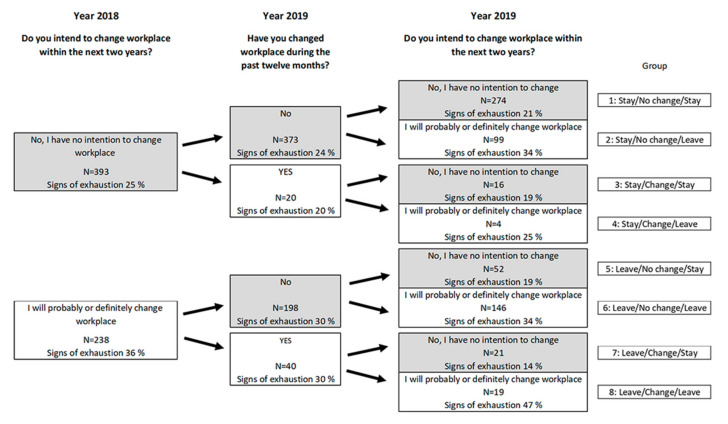
Longitudinal patterns of intended and actual change of workplace and its association with signs of exhaustion among 631 principals in compulsory schools.

#### 3.2.3. Group 7 (Leave/Change/Stay) and Group 8 (Leave/Change/Leave) 

The pairwise comparisons of occupational factors and signs of exhaustion in 2019 and 2018, within and between Group 7 (N = 21) and Group 8 (N = 19), are given in Appendix A.

## 4. Discussion

### 4.1. Principal Findings

A general finding based on both parts of the study is that most of the principals with the intention to stay at the present workplace reported better working conditions and a lower mean sum score on the exhaustion scale (i.e., KEDS), compared to those who had the intention to change workplace. 

The multivariate analyses indicated that the occupational factors that were most strongly associated with the intention to change workplace, were a demanding buffer-function between management and co-workers, demanding role conflicts, and a perception of high external expectations from parents. Furthermore, an increased score on the KEDS exhaustion scale was associated with the intention to change workplace. An intention to stay at the current workplace was most strongly associated with Supportive management and Supportive manager colleagues. 

In Part II of the study, the longitudinal analyses indicated that changes in the intention to stay at, or leave, the present workplace showed associations with changes in the working conditions and the level of exhaustion. For example, the principals in Group 2 (Stay/No change/Leave) had no intention to change workplace 2018, but they did have such an intention in 2019. At the same time, they reported an increased Buffer-function, increased Resource deficits, and less Supportive management, compared to 2018. The principals in Group 5 (Leave/No change/Stay) had the intention to change workplace in 2018, but did not report any actual change in 2019. In 2019, they no longer intended to change workplace and also reported reduced Role conflicts, reduced Container functions, less stressful external expectations from the superintendent and the immediate supervisor, and markedly improved scores on the exhaustion scale, compared to 2018. Thus, the present results point to the need for improved working conditions among principals, for their well-being and for a reduction in their turnover. The most important preventive measures seem to be improving managerial support, reducing the demanding Buffer-function between management and co-workers, and reducing Role conflicts.

### 4.2. Methodological Considerations

A strength of the study is that it involved principals from 277 out of 290 municipalities in Sweden, comprising principals in cities, villages, and schools of all sizes. In this respect, the present participants are likely to be fairly representative of the entire population of principals in Swedish compulsory schools. Whether the conditions reported can be extrapolated to the working conditions among principals in an international context is more uncertain, and probably mostly due to specific circumstances in various countries and cultural settings.

Another strength of the study is the use of multiple methods to address the research questions, i.e., cross-sectional analyses with repeated measurements, together with an analysis of the longitudinal patterns in subgroups of principals with varying intentions to stay, or leave, the workplace. The fact that similar results were found in both parts of the study strengthens its validity.

The present study included principals who participated in training programs for principals funded and arranged by the Swedish National Agency for Education, during the period 2008–2017. Thus, both principals who participated in the training programs before 2008 (most likely older than the average for the study sample), and those who still had not passed the training program (most likely younger than the average) were not included. However, since the study sample is representative of Swedish principals concerning age [31], this might be a minor problem. Another concern is that principals who are better prepared for the positions are supposed to be less stressed and stay longer at the workplace [2]. Thus, the present selection of only trained principals may have resulted in an underestimation of the signs of both exhaustion and the principal turnover.

In the original study sample of principals at all school-levels, 2317 out of the 9900 who were invited responded to the questionnaire, which generated a response rate of 23 percent. However, as the invitation list included principals who participated in the training program in 2008–2017, there may have been an unknown fraction of principals who had left the profession before the first survey in 2018, and thus did no longer belong to the target group. Among the 4640 principals who were successfully contacted, 50 percent responded to the questionnaire. Although similar low response rates are common in large-scale surveys [23,32], this is a limitation of the study. Another limitation is that all of the data were self-reported. Furthermore, although it is likely that adverse working conditions in many cases may have preceded the intention to change workplace, the limitations of the study design preclude a definite statement on causal relationships.

In Part II of the study, the described patterns of intended and actual changes of workplace resulted in eight different scenarios/groups. Because some of the scenarios were less common, some of the groups had a limited number of participants. This was the case in particular for Groups 3 and 4, and they were therefore omitted from further analyses. Additionally, the analyses within and between Groups 7 and 8 suffered from a low power, which limited the possibility to detect statistically significant differences, if any were present.

### 4.3. Signs of Exhaustion

Several previous studies have shown associations between work environment factors and burnout/signs of exhaustion (e.g., [26,33]), which most likely apply also to the principals in the present study. Thus, it is reasonable to assume that the associations between signs of exhaustion and the intention to change workplace, to a large extent, are mediated through demanding circumstances in the work environment.

### 4.4. Should I Stay at the Present Workplace…? 

A high score of Supportive management, i.e., that the principals trust that superiors, when needed, will help them solve problems and that they express a genuine interest for the principals’ work and leadership, was strongly associated with the intention to stay at the present workplace. As a leader, the principal is the one that is supposed to give support to the staff [34], but the principal, in turn, needs support and guidance from the superintendent or the school owner [14]. Furthermore, a high score of Supportive manager colleagues was associated with the intention to stay. Thus, the present results confirm previous findings that support from colleagues and managers belongs to the factors that are decisive for the will to continue working in the workplace [25,35]. That supportive leadership is an important work environment issue is also confirmed by its protective effect against exhaustion and burnout [19,20,26]. Of course, due to the nature of the data, low scores in the demanding organizational and social work environment factors presented below also represent factors that were associated with the intention to stay.

### 4.5. … Or Should I Go?

Among the occupational factors, the strongest association with the intention to change workplace was found for a high score of Buffer-function, i.e., that the principals perceived themselves to be squeezed between co-workers and higher levels in the organization, and that superiors expect them to be understanding and committed to accepting decisions that were bad for them and the organization, and to advocate for such decisions towards their subordinates [30]. The challenge of acting as a buffer function may be related to other factors that were associated with the intention to change workplace, i.e., low managerial support and stressful external expectations from the superintendent and immediate manager. Insufficient managerial support is often reflected as a lack of trust between principals and the actors in the local governance chain [14]. The Buffer-function may also be related to a lack of decision-making authority, (i.e., deficient influence, control, or autonomy to shape decisions and solutions in the school) which has been emphasized to be one of the most important reasons for principal turnover in a US context [2]. Additionally, Resource deficits, reflected by lacking resources due to decisions by superiors, politicians, or governmental authorities, can be closely related to the Buffer-function.

Burdensome Role conflicts among principals may have to do with frictions between administrative work tasks, organizational development, being a pedagogical leader, and taking part in daily activities in interactions with co-workers. Overall, this may result in an excessive workload. Thus, besides a possible perceived sense of inadequacy among principals, the association between Role conflicts and intention to change workplace may be influenced by the complexity of the job and the amount of time needed to complete all necessary activities [2].

A perception of stressful external expectations from parents was associated with the intention to change workplace. According to Leo et al. [14], most parents are supportive. However, among complaints from parents that were reported to the Swedish schools-inspectorate, a majority concerned children and students who did not receive the support they needed, according to their parents. Responding to complaints was perceived as a stressful and time-consuming task for the principals [14]. The present finding of a high score of stressful external expectations from the Swedish schools-inspectorate can most likely be, and largely is, related to their handling of complaints from parents [14].

The intention to change workplace was more frequent among younger principals than among older ones, and more frequent among assistant principals than in principals. This may partly be explained by young assistant principals accepting an interim and challenging period of hard work as a stepping-stone in order to display merit for a future position as full principal. 

Moreover, the dimension Supportive private life comprised questions on the opportunities to rest and to relax from work during leisure time. No statistically significant association was found in the bivariate analysis, but in the multivariate analyses, a high score turned out to be associated with the intention to change workplace. A possible explanation is that a supportive private life might give a principal the inner strength to decide to leave a workplace associated with difficult working conditions and declining well-being and health.

### 4.6. Possible Implications

Due to the fact that an unstable management function is associated with a high teacher turnover [1], which in turn may lead to poor results for the students, the situation for principals poses an important societal issue. In order to increase retention among principals, systematic efforts to establish improved working conditions for principals seem to be sorely needed. It is likely that a joint effort along the entire governance chain, i.e., at the national, municipal, and local level, intending to improve working conditions for principals, would be a fruitful endeavor.

## 5. Conclusions

A general finding was the clear association between the principals’ working conditions and their intention to stay at or leave their current workplace. Poor working conditions in terms of, primarily, a lack of support from management, the feeling of being squeezed between management and co-workers (Buffer-function), and demanding Role conflicts were associated with a consideration to change workplace. At the same time, principals who intended to change workplace reported more signs of exhaustion. 

## Figures and Tables

**Table 1 ijerph-18-05376-t001:** Characteristics among participants included in Part I of the study, regarding reported conditions at work in the years 2018 (N = 984) and 2019 (N = 884). The number of participants who responded to both questionnaires was 633.

	Categories/Scale	2018	2019
		N = 984	N = 884
School owner; N (%)	Municipality	853 (87)	762 (86)
	Other organization	131 (13)	122 (14)
Job title; N (%)	Principal	717 (73)	665 (75)
	Assistant principal	267 (27)	219 (25)
Seniority as principal; N (%)	<3 years	202 (21)	177 (20)
	3–10 years	554 (56)	676 (76)
	>10 years	228 (23)	31 (4)
Number of co-workers; N (%)	0–20	180 (18)	146 (17)
	21–30	273 (28)	259 (29)
	31–40	294 (30)	260 (29)
	41–130	235 (24)	219 (25)
Staff access/availability; N (%)	Full staff	571 (58)	578 (65)
	Very or somewhat understaffed	413 (42)	306 (35)
Number of students; N (%)	0–200	343 (35)	283 (32)
	201–400	466 (47)	446 (50)
	401–600	139 (14)	115 (13)
	601–1377	35 (4)	40 (5)
Overtime work; N (%)	Once a week or less often	164 (17)	177 (20)
	Every day or a few days/week	779 (79)	676/(76)
	No agreed working hours	41 (4)	31 (4)
Perceived physical working environment; N (%)	Adequate, good or very good	762 (77)	700 (79)
	Poor or very poor	222 (23)	184 (21)
**Demanding organizational and social work environment; mean (SD)**			
Resource deficits	Scale 1–5	3.5 (0.9)	3.5 (1.0)
Organizational Control	Scale 1–5	2.6 (0.9)	2.6 (0.9)
Role conflicts	Scale 1–5	3.8 (0.8)	3.7 (0.8)
Role demands	Scale 1–5	3.1 (0.8)	3.1 (0.8)
Group dynamics	Scale 1–5	2.3 (0.7)	2.3 (0.7)
Buffer-function	Scale 1–5	2.9 (0.9)	2.9 (0.9)
Co-workers	Scale 1–5	2.9 (0.7)	2.8 (0.7)
Container- function	Scale 1–5	3.4 (0.8)	3.3 (0.8)
**Supportive organizational and social work environment; mean (SD)**			
Supportive management	Scale 1–5	3.1 (1.1)	3.1 (1.1)
Cooperating with co-workers	Scale 1–5	4.3 (0.6)	4.3 (0.6)
Supportive manager colleagues	Scale 1–5	3.9 (1.0)	3.8 (1.0)
Supportive private life	Scale 1–5	3.8 (1.0)	3.9 (1.0)
Supportive organizational structures	Scale 1–5	3.6 (0.9)	3.6 (0.9)
**Stressful external expectations; N (%)** ^a^			
The National Agency of Education	Score ≥ 4	322 (33)	250 (28)
The Swedish schools-inspectorate	Score ≥ 4	599 (61)	519 (59)
School owner	Score ≥ 4	440 (45)	385 (44)
Super intendent	Score ≥ 4	312 (32)	260 (29)
Immediate supervisor	Score ≥ 4	288 (29)	257 (29)
Co-workers	Score ≥ 4	484 (49)	374 (42)
Parents	Score ≥ 4	544 (55)	505 (57)
Students	Score ≥ 4	181 (18)	166 (19)
**Karolinska Exhaustion disorders scale (KEDS)**			
Mean score (SD)	Score 0–54	14 (8)	14 (9)
Possible exhaustion disorder; N (%)	Score ≥ 19 points	290 (29)	232 (26)
Change of workplace in the past five years; N (%)	None	385 (39)	n.a.
	Once	393 (40)	
	Twice	121 (12)	
	Three times or more	85 (9)	
Change of workplace in the past twelve months; N (%)	None	n.a.	797 (90)
	Once		82 (9)
	Twice		4 (0.5)
	Three times or more		1 (0.1)
Intention to change workplace within two years; N (%)	Yes, definitely	98 (10)	96 (11)
	Yes, probably	304 (31)	271 (31)
	No	559 (57)	495 (56)
	I will retire	23 (2)	22 (2)

^a^ The seven-point scales (0–6) were dichotomized into low (<4 points) and high (≥4 points). Bold faces may help the reader to identify the different categories of dimensions.

**Table 2 ijerph-18-05376-t002:** Cross-sectional bivariate and multivariate binary logistic regression analysis between occupational factors, gender age, and signs of exhaustion and the dichotomous outcome of intention to change workplace (no vs. yes, probably/yes, definitely) in the total study sample, taking repeated assessments in the years 2018 (N = 984) and 2019 (N = 884) into account. Odds ratio (OR) and 95% confidence intervals (CI) are presented, which in continuous variables (GMSI and KEDS) are associated with a one-unit increase/decrease in mean score on the scale. Results in bold face are statistically significant.

Independent Factors		Bivariate	Multivariate ^a^
	Categories/Scale	OR (CI 95%)	OR (CI 95%)
Gender	Female	1	-
	Male	1.15 (0.91–1.46)	
Age	years	**0.97 (0.96–0.99)**	**0.97 (0.96–0.99)**
School owner	Municipality	1	-
	Other organization	0.93 (0.67–1.28)	
Job title	Principal	**1**	**1**
	Assistant principal	**1.39 (1.09–1.76)**	**1.43 (1.10–1.84)**
Seniority as principal	<3 years	1	-
	3–10 years	1.19 (0.89–1.59)	
>10 years	1.18 (0.84–1.66)
Number of co-workers	0–20	1	-
	21–30	1.25 (0.92–1.71)	
	31–40	1.15 (0.84–1.57)	
	41–130	1.22 (0.88–1.68)	
Staff access/availability	Full staff	**1**	1
	Very or somewhat understaffed	**1.31 (1.07–1.60)**	0.94 (0.76–1.16)
Number of students	0–200	1	-
	201–400	1.09 (0.87–1.37)	
	401–600	1.08 (0.78–1.49)	
	601–1377	1.31 (0.73–2.33)	
Overtime	Once a week or less often	1	-
	Every day or a few days/week	0.98 (0.75–1.27)	
	No agreed working hours	0.88 (0.49–1.58)	
Physical working environment	Adequate, good or very good	1	1
	Poor or very poor	**1.54 (1.22–1.95)**	1.10 (0.85–1.42)
**Demanding organizational and social work environment**			
Resource deficits	Score 1–5	**1.21 (1.09–1.34)**	0.93 (0.82–1.07)
Organizational Control	Score 1–5	**1.48 (1.31–1.67)**	1.00 (0.85–1.18)
Role conflicts	Score 1–5	**1.57 (1.37–1.80)**	**1.22 (1.03–1.46)**
Role demands	Score 1–5	**1.41 (1.23–1.61)**	0.85 (0.70–1.03)
Group dynamics	Score 1–5	**1.54 (1.34–1.78)**	1.13 (0.93–1.36)
Buffer-function	Score 1–5	**1.61 (1.43–1.81)**	**1.27 (1.08–1.50)**
Co-workers	Score 1–5	**1.45 (1.26–1.68)**	1.14 (0.95–1.37)
Container-function	Score 1–5	**1.35 (1.19–1.54)**	0.86 (0.72–1.02)
**Supportive organizational and social work environment**			
Supportive management	Score 1–5	**0.67 (0.61–0.74)**	**0.79 (0.71–0.89)**
Cooperating with co-workers	Score 1–5	**0.69 (0.59–0.81)**	0.96 (0.79–1.17)
Supportive manager colleagues	Score 1–5	**0.70 (0.63–0.77)**	**0.83 (0.74–0.94)**
Supportive private life	Score 1–5	0.92 (0.83–1.01)	**1.17 (1.04–1.31)**
Supportive organizational structures	Score 1–5	**0.71 (0.63–0.79)**	0.95 (0.83–1.09)
**Stressful external expectations**			
The National Agency of Education	Score < 4	1	
	≥4	0.90 (0.73–1.12)	
The Swedish schools-inspectorate	Score < 4	1	
	≥4	0.98 (0.80–1.19)	
School owner	Score < 4	**1**	-
	≥4	**1.40 (1.15–1.70)**	
Super Intendent	Score < 4	**1**	-
	≥4	**1.32 (1.07–1.63)**	
Immediate supervisor	Score < 4	**1**	-
	≥4	**1.73 (1.39–2.14)**	
Co-workers	Score < 4	**1**	-
	≥4	**1.64 (1.35–2.00)**	
Parents	Score < 4	1	**1**
	≥4	**1.57 (1.28–1.91)**	**1.32 (1.05–1.65)**
Students	Score < 4	1	1
	≥4	1.24 (0.97–1.59)	1.02 (0.77–1.34)
Karolinska Exhaustion disorder scale (KEDS)	Score 0–54	**1.05 (1.04–1.06)**	**1.03 (1.02–1.05)**

^a^ The variables stressful external expectations from school owner, super intendent, immediate supervisor, and co-workers were omitted from the multivariate analyses due to a conceptual overlap with the demanding and supporting dimensions in GMSI. Bold faces are remained to indicate which factors that are statistically significant (as stated in the Table text).

**Table 3 ijerph-18-05376-t003:** Changes of reported exposures (2018 vs. 2019) within Group 1 (Stay/No change/Stay) and within Group 2 (Stay/No change/Leave) (Figure 1); and comparison of exposures between Group 1 and Group 2.

		Group 1 (Stay/No Change/Stay) N = 274			Group 2 (Stay/No Change/Leave) N = 99			Difference between Group 1 and Group 2 *p*-Value ^A^	
Independent Factors	Categories/Scale	2018	2019	2018 vs. 2019; *p*-Value ^B^	2018	2019	2018 vs. 2019; *p*-Value ^B^	2018	2019
Gender (2018); N (%)	Female	207 (76)			68 (69)			0.19	
	Male	67 (24)			31 (31)				
Age (2018); mean (SD)	years	50 (7)			47 (7)			<0.001	
School owner; N (%)	Municipality	229 (84)	229 (84)	1.00	91 (92)	90 (91)	1.00	0.04	0.09
	Other organization	45 (16)	45 (16)		8 (8)	9 (9)			
Job title; N (%)	Principal	205 (76)	207 (76)	0.51	72 (73)	72 (73)	1.00	0.69	0.59
	Assistant principal	69 (25)	66 (24)		27 (27)	27 (27)			
	Other title ^C^	-	1 (0.4)		-	0			
Number of co-workers; mean (SD)		34 (16)	34 (15)	0.77	34 (12)	33 (12)	0.56	0.98	0.93
Staff access/availability	Full staff	166 (61)	192 (70)	<0.01	56 (57)	64 (65)	0.22	0.55	0.32
	Very or somewhat understaffed	108 (39)	82 (30)		43 (43)	35 (35)			
Number of students; mean (SD)		277 (166)	286 (166)	<0.01	270 (136)	252 (136)	0.52	0.65	0.05
Overtime	Once a week or less often	44 (16)	60 (22)	<0.01	11 (11)	19 (19)	0.09	0.25	0.67
	Every day or a few days/week	215 (78)	206 (75)		85 (86)	76 (77)			
	No agreed working hours ^D^	15 (5)	8 (3)		3 (3)	4 (4)			
Physical working environment; N (%)	Adequate, good or very good	229 (84)	222 (81)	0.39	72 (73)	73 (74)	1.00	0.03	0.15
	Poor or very poor	45 (16)	52 (19)		27 (27)	26 (26)			
**Demanding organizational and social work environment; mean (SD) **									
Resource deficits	Score 1–5	3.4 (0.9)	3.4 (1.0)	0.33	3.6 (0.9)	3.8 (0.9)	0.01	0.21	<0.001
Organizational Control	Score 1–5	2.5 (0.9)	2.4 (0.9)	0.66	2.6 (0.8)	2.7 (0.8)	0.23	0.15	0.02
Role conflicts	Score 1–5	3.6 (0.8)	3.5 (0.9)	<0.01	3.9 (0.8)	3.8 (0.8)	0.35	<0.01	<0.01
Role demands	Score 1–5	3.0 (0.8)	3.0 (0.8)	0.41	3.2 (0.8)	3.2 (0.8)	0.17	0.06	0.02
Group dynamics	Score 1–5	2.2 (0.7)	2.1 (0.7)	0.02	2.4 (0.7)	2.4 (0.7)	0.98	0.02	<0.001
Buffer-function	Score 1–5	2.7 (0.9)	2.7 (0.9)	0.87	2.9 (0.9)	3.2 (0.9)	0.001	0.05	<0.001
Co-workers	Score 1–5	2.8 (0.7)	2.7 (0.7)	0.19	2.9 (0.7)	2.9 (0.7)	0.80	0.05	0.03
Container- function	Score 1–5	3.3 (0.8)	3.2 (0.8)	0.02	3.5 (0.7)	3.4 (0.9)	0.12	0.04	0.04
**Supportive organizational and social work environment; mean (SD)**									
Supportive management	Score 1–5	3.4 (1.1)	3.4 (1.1)	0.44	**3.1** (**1.1**)	**2.9** (**1.2**)	**0.01**	**0.03**	**<0.001**
Cooperating with co-workers	Score 1–5	4.4 (0.6)	4.4 (0.6)	0.55	4.2 (0.7)	4.2 (0.7)	0.88	0.06	0.06
Supportive manager colleagues	Score 1–5	**4.0** (**0.9**)	**3.9** (**1.0**)	**0.05**	3.9 (1.0)	3.8 (0.9)	0.39	0.33	0.26
Supportive private life	Score 1–5	3.8 (1.0)	3.9 (1.0)	0.18	3.6 (1.1)	3.7 (1.0)	0.40	0.19	0.18
Supportive organizational structures	Score 1–5	**3.7** (**0.9**)	**3.8** (**0.9**)	**0.03**	3.6 (0.9)	3.5 (0.9)	0.52	0.24	**<0.01**
**Stressful external expectations; mean (SD)**									
The National Agency of Education	Score 1–7	3.5 (1.8)	3.3 (1.8)	0.27	3.5 (1.8)	3.3 (1.9)	0.40	0.81	0.90
The Swedish schools-inspectorate	Score 1–7	4.9 (1.8)	4.7 (1.9)	0.19	4.9 (1.8)	4.6 (2.0)	0.06	0.85	0.45
School owner	Score 1–7	4.0 (1.5)	3.9 (1.7)	0.32	4.2 (1.6)	4.3 (1.6)	0.48	0.18	**0.03**
Super Intendent	Score 1–7	3.4 (1.6)	3.3 (1.7)	0.17	3.8 (1.7)	3.6 (1.7)	0.49	0.11	0.11
Immediate supervisor	Score 1–7	3.2 (1.6)	3.1 (1.7)	0.19	3.4 (1.6)	3.7 (1.6)	0.26	0.23	**<0.01**
Co-workers	Score 1–7	4.1 (1.4)	3.8 (1.5)	<0.01	4.5 (1.6)	4.4 (1.6)	0.62	<0.01	<0.001
Parents	Score 1–7	4.4 (1.7)	4.4 (1.7)	0.90	4.8 (1.6)	4.7 (1.5)	0.49	0.05	0.16
Students	Score 1–7	2.9 (1.5)	2.9 (1.6)	0.89	3.0 (1.6)	3.1 (1.8)	0.26	0.53	0.42
**Karolinska Exhaustion Disorder Scale**									
Mean score (SD)	Score 0–54	13.1 (7.7)	12.7 (8.0)	0.08	15.0 (7.9)	15.9 (8.0)	0.18	0.04	<0.001
Possible exhaustion disorder; N (%)	Score ≥ 19 points	66 (24)	57 (21)	0.21	29 (29)	34 (34)	0.40	0.35	<0.01

^A.^ The Mann–Whitney U-test was used for continuous variables, and Fisher’s exact test was used for dichotomous variables. ^B.^ The Wilcoxon Signed Rank Test was used for continuous variables, and McNemar’s test for dichotomous variables. ^C.^ One person with “other title” in 2019 was excluded from the statistical analysis. ^D.^ Participants with “No agreed working hours” were excluded from the statistical analysis.

**Table 4 ijerph-18-05376-t004:** Change of reported exposures in the year 2018 vs. in the year 2019, within and between Group 5 (Leave/No change/Stay) and Group 6 (Leave/No change/Leave) (Figure 1).

		Group 5			Group 6			Difference between Group 5 and Group 6	
		N = 52			N = 146			*p*-Value ^A^	
Independent factors	Categories/Scale	2018	2019	2018 vs. 2019; *p*-Value ^B^	2018	2019	2018 vs. 2019; *p*-Value ^B^	2018	2019
Gender (2018); N (%)	Female	42 (81)			113 (77)			0.70	
	Male	10 (19)			33 (23)				
Age (2018); mean (SD)	years	51 (6)			48 (7)			**0.02**	
School owner; N (%)	Municipality	45 (87)	45 (87)	1.00	123 (84)	123 (84)	1.00	0.82	0.82
	Other organisation	7 (13)	7 (13)		23 (16)	23 (16)			
Job title; N (%)	Principal	38 (73)	41 (79)	0.12	95 (65)	97 (66)	0.62	0.31	0.08
	Assistant principal	14 (27)	10 (19)		51 (35)	49 (34)			
	Other title ^C^	-	1 (1.9)		-	0			
Number of co-workers; mean (SD)		33(12)	34 (16)	0.95	35 (16)	34 (17)	0.60	0.62	0.92
Staff access/availability	Full staff	28 (54)	31 (60)	0.65	90 (62)	89 (61)	1.00	0.33	0.87
	Very or somewhat understaffed	24 (46)	21 (40)		56 (38)	57 (39)			
Number of students; mean (SD)		256 (136)	287 (184)	0.07	303 (197)	307 (202)	0.12	0.19	0.49
Overtime	Once a week or less often	7 (13)	12 (23)	0.06	29 (20)	32 (22)	0.72	0.40	1.00
	Every day or a few days/week	43 (83)	39 (75)		111 (76)	107 (73)			
	No agreed working hours ^D^	2 (4)	1 (2)		6 (4)	7(5)			
Physical working environment; N (%)	Adequate, good or very good	40 (77)	42 (81)	0.77	114 (78)	110 (75)	0.56	0.85	0.57
	Poor or very poor	12 (23)	10 (19)		32 (22)	36 (25)			
**Demanding organisational and** **social work environment; mean (SD)**									
Resource deficits	Score 1-5	3.6 (1.0)	3.6 (1.0)	0.62	3.5 (1.0)	3.6 (1.0)	0.35	0.33	0.64
Organisational Control	Score 1-5	2.8 (0.8)	2.8 (0.9)	0.83	2.7 (0.8)	2.7 (0.9)	0.59	0.28	0.52
Role conflicts	Score 1-5	**4.0 (0.8)**	**3.6 (0.8)**	**0.001**	3.9 (0.8)	3.9 (0.8)	0.79	0.72	**0.02**
Role demands	Score 1-5	3.2 (0.8)	3.0 (0.7)	0.07	3.2 (0.8 )	3.2 (0.8 )	0.39	0.76	0.11
Group dynamics	Score 1-5	2.3 (0.8)	2.3 (0.8)	0.95	2.4 (0.8)	2.4 (0.7)	0.42	0.52	0.92
Buffer-function	Score 1-5	3.1 (0.9)	3.1 (1.0)	0.52	3.1 (0.9)	3.1 (0.9)	0.48	0.96	0.40
Co-workers	Score 1-5	2.8 (0.8)	2.6 (0.8)	0.07	3.0 (0.7)	3.0 (0.7)	0.34	0.22	**0.02**
Container- function	Score 1-5	3.6 (0.8)	3.3 (0.9)	**0.02**	3.5 (0.8)	3.4 (0.8)	0.45	0.21	0.39
**Supportive organisational and** **social work environment; mean (SD)**									
Supportive management	Score 1–5	3.0 (1.1)	3.0 (1.0)	0.69	2.9 (1.0)	2.9 (1.1)	0.81	0.74	0.56
Cooperating with co-workers	Score 1–5	4.2 (0.6)	4.2 (0.6)	0.94	4.2 (0.7)	4.2 (0.6)	0.45	0.63	0.65
Supportive manager colleagues	Score 1–5	3.8 (0.8)	3.9 (1.0)	0.87	3.6 (1.1)	3.6 (1.0)	0.67	0.35	0.20
Supportive private life	Score 1–5	3.8 (1.1)	3.9 (1.0)	0.16	3.8 (1.1)	3.9 (1.0)	0.74	0.88	0.84
Supportive organisational structures	Score 1–5	3.5 (0.9)	3.8 (0.9)	0.07	3.6 (1.0)	3.5 (1.0)	**0.03**	0.29	0.17
**Stressful external** **expectations; mean (SD)**									
The National Agency of Education	Score 1–7	3.7 (1.9)	3.3 (1.8)	0.16	3.4 (1.8)	3.5 (1.8)	0.41	0.40	0.44
The Swedish schools-inspectorate	Score 1–7	4.9 (2.0)	5.0 (1.8)	0.95	4.8 (1.8)	4.9 (1.8)	0.69	0.49	0.77
School owner	Score 1–7	4.3 (1.8)	4.1 (1.6)	0.47	4.4 (1.6)	4.4 (1.6)	0.81	0.80	0.35
Super intendent	Score 1–7	4.1 (1.6)	3.4 (1.7)	**0.007**	3.8 (1.6)	3.8 (1.7)	0.82	0.22	0.18
Immediate supervisor	Score 1–7	3.8 (1.8)	3.2 (1.7)	**0.02**	3.8 (1.8)	3.8 (1.8)	0.88	0.99	**0.04**
Co-workers	Score 1–7	4.2 (1.5)	3.8 (1.6)	0.13	4.7 (1.5)	4.4 (1.5)	**0.01**	0.06	**0.04**
Parents	Score 1–7	4.9 (1.5)	4.8 (1.7)	0.72	5.0 (1.6)	5.0 (1.6)	0.90	0.62	0.49
Students	Score 1–7	2.9 (1.5)	3.0 (1.8)	0.80	3.3 (1.6)	3.3 (1.7)	0.19	0.19	0.19
**Karolinska Exhaustion Disorder Scale**									
Mean score (SD)	Score 0–54	15 (7.2)	12 (7.6)	**<0.001**	16 (8.9)	16 (9.1)	0.51	0.49	**0.006**
Possible exhaustion disorder; N (%)	Score ≥ 19 points	15 (29)	10 (19)	0.18	55 (38)	50 (34)	0.44	0.31	0.05

^A.^ The Mann–Whitney U-test was used for continuous variables, and Fisher’s exact test was used for dichotomous variables. ^B.^ The Wilcoxon Signed Rank Test was used for continuous variables, and McNemar’s test for dichotomous variables. ^C.^ One person with “other title” in 2019 was excluded from the statistical analysis. ^D.^ Participants with “No agreed working hours” were excluded from the statistical analysis.

## Data Availability

Consistent with the study protocol approved by the Regional Ethical Review Board, anonymized data are stored locally at the Division of Occupational and Environmental Medicine, Lund University, Lund, Sweden. In accordance with the ethical approval, crude data are not to be published on the internet. Access to data will be granted to eligible researchers wanting to audit our research. Requests should be directed to the corresponding author.

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
