# Peer review of "Should I Stay or Should I Go? Associations between Occupational Factors, Signs of Exhaustion, and the Intention to Change Workplace among Swedish Principals"

_ijerph, 2021, doi:10.3390/ijerph18105376_

Round 1

Reviewer 1 Report

Background

  1. The knowledge of working conditions and exhaustion among principals could be explained more profoundly. It is mentioned that more research on teachers has been conducted, however you also write “Anyhow, there are several reports of complex and demanding working conditions among principals [12-15]”. Are there no studies or reviews of the working conditions among principals or other relevant managers. If not, please clarify that. Otherwise describe further what is already known.

Materials and Methods

  1. Figure 1 is referred to at the end of the paragraph “participants”, however it seems misleading to refer to Figure 1in relation to that section. The use of a flow chart to depict the study sample would be helpful. Anyhow, removing the referral to Figure 1 on line number 144 is suggested.
  2. Intention to leave is described as the outcome variable in Part I, however “Intention to stay” is also reported as an outcome variable. Thus, describe how “Intention to stay” was generated. Further, this is confusing throughout the paper as you refer to staying or leaving (go) in the title and in the discussion, however staying is not mentioned in the aim, nor outlined as an outcome variable. On line 497 you touch upon this, however it would be beneficial if this was clarified earlier in the paper.
  3. In Part II it is not clear in what way step 2 (changed workplace in last 12 months) contributes to the study design. This could be due to my ignorance, however it seems like other groups, with higher N/power could have been produced without this step. Please motivate in the paper why this step is necessary.  
  4. It is difficult to follow the different groups (1-8), perhaps naming them would make it easier for the reader to follow.

Results

  1. 1.2. Multi-variate analyses of associations between occupational factors, age, and signs of exhaus- tion, and the intention to change workplace.

This section is difficult to follow. A suggestion for clarification is to only describe the changes in relation to the bivariate analysis, which factors remained significant, and which did not.

  1. You write: The strongest association with an intention to change workplace was found for a high sum-score on the exhaustion scale (KEDS). This comes as a bit of surprise as the OR in table 2 reports KEDS (OR 1.03; CI 1.02 – 1.05). Perhaps I have missed something, however this association does not come across as the strongest in relation to the outcome and in comparison, to other variables.
  2. In table 2 you report on OR:s, however for the factors deriving from the GMSI and for the exhaustion scale no reference group is pointed out due to the nature of the scales. However, to help the reader to understand how to interpret the results, a clarification of how the results should be interpreted is suggested. Another example of this: Furthermore, an increased score on the KEDS exhaustion scale was associated with the intention to change workplace (OR 1.03; CI 1.02 – 1.05; associated with a one unit increase on the scale).

Discussion

  1. The paragraph on line 465 about omitted participants due to retirement seems out of context. It is unclear how the statements in the paragraph are related to the analysis of the study. If the exclusion of this group is seen as a limitation, or consideration, please state that and describe in what way.

Minor changes

  1. Sometimes physical working environment is used, and sometimes physical work environment is used, check and revise.
  2. Line 300. In the table caption Bivariate is used and within the table Univariate is used, please align.
  3. Line 484. The word work is written twice (work workplace).
  4. Headline Part II on line 220 needs to be evolved. Describe to the reader what this paragraph is about.
  5. Line 257 the phrase: in the second part of the study is used, which is assumed to refer to Part II. The use of the same phrase (Part II) throughout the study is recommended.
  6. It is stated: “Were a demanding buffer function between management and co-workers (OR 1.27; CI 1.08 – 1.50)”. No numbers are needed in the discussion part, please delete.

Reviewer 2 Report

I want to congratulate the authors, it is a nice piece of research. However, the piece lacks a proper narrative and it is rather easy to get lost in the presentation of the numbers and the statistical analyses.

Halfway through the paper, I got lost and had to go back. In short, It could be greatly improved should you spend more time designing the way you present your paper. 

My suggestion is that you take all the analysis out, for a moment, all the numbers out and then see what you get, then start rebuilding from there incorporating what is necessary, otherwise, as I said, the paper easily becomes dense and uninteresting.  

I know how to interpret things and I assume your reader will too, so do not give everything and try to explain everything at once.

Best of luck

Round 2

Reviewer 1 Report

Dear authors,

Thank you for an interesting paper. After the revision I accept the paper in present form.

Author Response

Thank you very much!

Reviewer 2 Report

I appreciate the authors addressing my concerns. I still believe it is crowded with data that in my opinio0n is unnecessary to include, a better narrative would be desirable. Again, it is not that the research is wrong, it is that an interesting topic and what could be a highly cited article is thrown to waste because of the bad structure of the paper. It is hard to follow and very easy to get lost.

Author Response

Thank you. As this type of research includes statistical analysis of the collected data, we are not sure of which specific improvements the reviewer ask for. Since no specific comments or suggestions has been provided, no further changes were made to the manuscript.